# Curvature Clues: Decoding Deep Learning Privacy with Input Loss Curvature

**Deepak Ravikumar    Efstathia Soufleri    Kaushik Roy**
Department of Electrical and Computer Engineering
Purdue University
West Lafayette, IN 47907
`{dravikum, esoufler, kaushik}@purdue.edu`

## Abstract

In this paper, we explore the properties of loss curvature with respect to input data in deep neural networks. Curvature of loss with respect to input (termed input loss curvature) is the trace of the Hessian of the loss with respect to the input. We investigate how input loss curvature varies between train and test sets, and its implications for train-test distinguishability. We develop a theoretical framework that derives an upper bound on the train-test distinguishability based on privacy and the size of the training set. This novel insight fuels the development of a new black box membership inference attack utilizing input loss curvature. We validate our theoretical findings through experiments in computer vision classification tasks, demonstrating that input loss curvature surpasses existing methods in membership inference effectiveness. Our analysis highlights how the performance of membership inference attack (MIA) methods varies with the size of the training set, showing that curvature-based MIA outperforms other methods on sufficiently large datasets. This condition is often met by real datasets, as demonstrated by our results on CIFAR10, CIFAR100, and ImageNet. These findings not only advance our understanding of deep neural network behavior but also improve the ability to test privacy-preserving techniques in machine learning.[1] [2]

## 1 Introduction

Deep neural networks are being increasingly trained on sensitive datasets; thus ensuring the privacy of these models is paramount. Membership inference attacks (MIA) have become the standard approach to test a model's privacy [Murakonda and Shokri, 2020]. These attacks take a trained model and aim to identify if a given example was used in its training. Recent work has linked curvature of loss with respect to input with memorization [Garg et al., 2024] and differential privacy [Dwork et al., 2006, Ravikumar et al., 2024]. Inspired by this line of research, we investigate the properties of input loss curvature and leverage our insights to develop a new membership inference attack.

Curvature of loss with respect to input (termed input loss curvature) is defined as the trace of the Hessian of loss with respect to the input [Moosavi-Dezfooli et al., 2019, Garg and Roy, 2023]. Prior works that study loss curvature have focused on two lines of research. The first line of research has focused on studying the loss curvature with respect to the weights of the deep neural net [Keskar et al., 2017, Wu et al., 2020, Jiang et al., 2020, Foret et al., 2021, Kwon et al., 2021, Andriushchenko and Flammarion, 2022] to better understand generalization. The second line of research studied the loss curvature with respect to the input (i.e. data) for gaining insight into adversarial robustness

---

[1]Code available at `https://github.com/DeepakTatachar/Curvature-Clues`

[2]Project website with models and other assets is available at `https://engineering.purdue.edu/NRL/projects/curvature-clues`

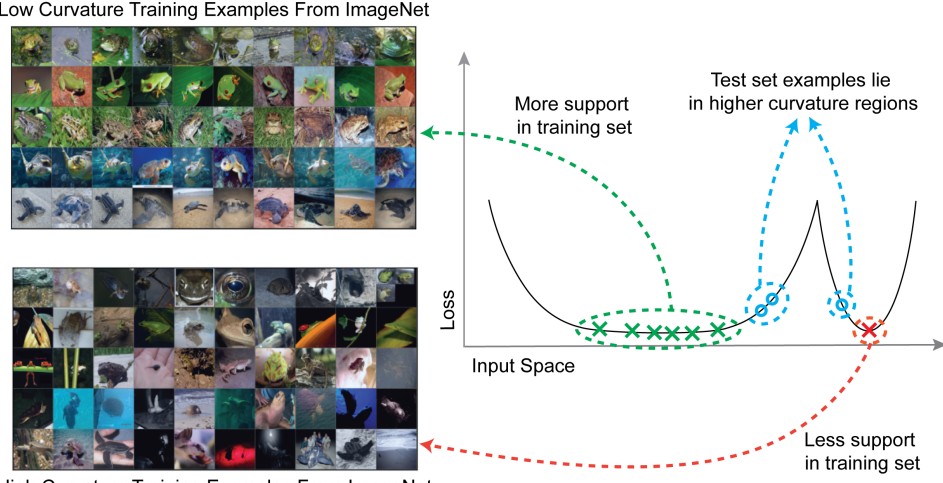

Figure 1: Visualizing low and high input curvature samples from a ResNet50 trained on ImageNet. Low input curvature training set images are prototypical and have lots of support in the trainset, while high input curvature train set examples have less support and are atypical. Test set examples lie around the training set images in higher curvature regions.

[Moosavi-Dezfooli et al., 2019, Fawzi et al., 2018], coresets [Garg and Roy, 2023] and memorization [Garg et al., 2024, Ravikumar et al., 2024], mainly focusing on its properties on the train set. Thus, there is a gap in our understanding of input loss curvature on unseen test examples.

Input loss curvature on the train set captures the prototypicality of an image. This is visualized in Figure 1 (left) which shows high curvature examples from ImagetNet [Russakovsky et al., 2015] train set. Figure 1 (right) builds intuition as to why this is the case – a sample with lots of support in the dataset is more likely to lie in low curvature regions while atypical/non-prototypical examples have less support in the training set and thus lie in higher curvature regions [Garg et al., 2024, Ravikumar et al., 2024]. With this established, we make a novel observation that on average the input loss curvature scores for test set examples are higher than train set examples. This is because test samples were not optimized for, hence they lie slightly off the flat minima, in regions of higher input curvature (also visualized in Figure 1). We leverage this insight and develop a theoretical framework that focuses on the distinguishability of train-test input loss curvature scores. Studying this distinguishability leads us to understand performance bounds of membership inference attacks [Shokri et al., 2017, Sablayrolles et al., 2019, Song and Mittal, 2021, Carlini et al., 2022].

Our theoretical analysis obtains an upper bound on the KL divergence between train-test input loss curvature scores. This provides a ceiling for membership inference attack performance. The analysis also reveals that the upper bound when using input loss curvature scores is dependent on the number of training set examples and the differential privacy parameter $\epsilon$. This insight helps us understand the conditions under which input loss curvature can be more effective at detecting train vs. test examples. Conditions such as the number of training examples and the privacy guarantees of the model.

To test our theoretical results we perform black-box (setting where adversaries have access only to model outputs) membership inference attack (MIA) using input loss curvature scores. However, input loss curvature score calculation needs access to model parameters (which are not available in a black box setting). To resolve this issue we propose using a zero-order input loss curvature estimation. Zero-order estimation can calculate input loss curvature without needing access to model parameters. Our input loss curvature-based black-box membership inference attack results show that real datasets contain enough training examples for curvature scores to outperform current state-of-the-art membership inference methods. Our results show that curvature based MIA outperforms prior state-of-the-art techniques on 10% or larger subsets of CIFAR100 dataset (about 5000 sample training set).

In summary, our contributions are as follows:

- Theoretical Foundation: We provide theoretical analysis to understand the train-test distinguishability with input loss curvature scores, demonstrating that input loss curvature is more

sensitive and hence better at detecting train vs. test set examples than current state-of-the-art membership inference attacks.

- Adapting Theory To Practice: We propose using zero order input loss curvature estimation to enable black box membership inference attack using input loss curvature scores.

- Better Attack: We conduct experiments to validate our theoretical results. Specifically, we show that input loss curvature enables more effective black box membership inference attacks when compared to existing state-of-the-art techniques.

## 2   Related Work

**Membership Inference Attacks** are used as a tool to test privacy [Murakonda and Shokri, 2020]. These attacks aim to identify if a particular data point was included in a model's training dataset [Shokri et al., 2017]. Existing techniques often leverage model outputs like loss values [Shokri et al., 2017, Yeom et al., 2018, Sablayrolles et al., 2019], confidence scores [Carlini et al., 2022] or modified entropy [Song and Mittal, 2021]. Shokri et al. [2017] proposed the used of shadow models which are auxiliary models trained on subsets of the target model's data to aid in inference. Several modifications to this approach have been proposed. One important addition is the focus on example hardness, where the authors Sablayrolles et al. [2019] proposed an attack that scaled the loss using per example hardness threshold which is estimated by training shadow models. Watson et al. [2022] proposed a similar approach but in the offline case where they calibrated the example hardness using the average loss of shadow models not trained on the target example. Ye et al. [2022] models various attacks into four categories and performs differential analysis to explain the gaps between them. Both Ye et al. [2022] and Long et al. [2020] consider the entire loss distribution of samples that are not in the training set. However, they face challenges in extrapolating to low false positive rates (FPRs). To address this issue Carlini et al. [2022] propose using a parametric model along with shadow models to improve performance. Orthogonal to these approaches Choquette-Choo et al. [2021] suggests the use of input augmentations during evaluation to improve the performance of the attacks. Similarly Jayaraman et al. [2021] propose the MERLIN attack, which queries the target model multiple times on a sample perturbed with Gaussian noise.

**Input Loss Curvature** is defined as the trace of the Hessian of loss with respect to the input [Moosavi-Dezfooli et al., 2019, Garg and Roy, 2023]. The aim is to measure the sensitivity of the deep neural network to a specific input. In general, loss curvature with respect to the parameters of deep neural nets has received lots of attention [Keskar et al., 2017, Wu et al., 2020, Jiang et al., 2020, Foret et al., 2021, Kwon et al., 2021, Andriushchenko and Flammarion, 2022], specifically due to its role in characterizing the sharpness of the learning objective which is closely connected to generalization. However, loss curvature with respect to the input data has received much less attention. It has been studied in the context of adversarial robustness [Fawzi et al., 2018, Moosavi-Dezfooli et al., 2019], coresets [Garg and Roy, 2023]. It has recently been linked with memorization [Garg et al., 2024, Ravikumar et al., 2024] and privacy [Ravikumar et al., 2024]. The authors in Moosavi-Dezfooli et al. [2019] showed that adversarial training decreases the input loss curvature and provided a theoretical link between robustness and curvature. In an orthogonal direction, Garg and Roy [2023] proposed the use of low input loss curvature examples as training dataset sets called coresets, which they showed to be more data-efficient. However, all of these works have focused on input loss curvature on the trainset. In this paper we focus on input loss curvature and its behavior on test or unseen examples to understand and improve the performance of membership inference attacks. Before we discuss our contributions, we present a few preliminaries, notation, and background needed for this paper

## 3   Notation and Background

**Notation.** Let us consider a supervised learning problem, where the goal is to learn a mapping from an input space $\mathcal{X} \subset \mathbb{R}^d$ to an output space $\mathcal{Y} \subset \mathbb{R}$. The learning is performed using a randomized algorithm $\mathcal{A}$ on a training set $S$. Note that a randomized algorithm employs a degree of randomness as a part of its logic. The training set $S$ contains $m$ elements denoted as $z_1, \cdots, z_m$. Each element $z_i = (x_i, y_i)$ is drawn from an unknown distribution $\mathcal{D}$, where $z_i \in \mathcal{Z}, x_i \in \mathcal{X}, y_i \in \mathcal{Y}$ and $\mathcal{Z} = \mathcal{X} \times \mathcal{Y}$. Thus, we define the training set $S \in \mathcal{Z}^m$ as $S = \{z_1, \cdots, z_m\}$. Another related concept is that of adjacent datasets, which are obtained when the set's $i^{th}$ element is removed and

defined as

$$S^{\backslash i} = \{z_1, \cdots, z_{i-1}, z_{i+1}, \cdots, z_m\}.$$

Additionally, the concept of adjacent (or neighboring) datasets is linked to the distance between datasets. The distance between two datasets $S$ and $S'$, denoted by $\|S - S'\|_1$, measures the number of samples that differ between them. The notation $\|S\|_1$ represents the size of a dataset $S$. When a randomized learning algorithm $\mathcal{A}$ is applied to a dataset $S$, it produces a hypothesis denoted by $h_S^\phi = \mathcal{A}(\phi, S)$, where $\phi \sim \Phi$ is the random variable associated with the algorithm's randomness. A cost function $c : \mathcal{Y} \times \mathcal{Y} \to \mathbb{R}^+$ is used to measure the hypothesis's performance. The cost of the hypothesis $h$ at a sample $z_i$ is also referred to as the loss $\ell$ at $z_i$, defined as $\ell(h, z_i) = c(h(x_i), y_i)$. The performance of a hypothesis $h$ is measured using risk $R(h) = \mathbb{E}_{z_i}[\ell(h, z_i)]$ and approximated using empirical risk $R_{emp}(h) = (1/m) \sum_{i=1}^m \ell(h, z_i)$.

**Differential Privacy.** Introduced by Dwork et al. [2006], differential privacy is defined as follows: A randomized algorithm $\mathcal{A}$ with domain $\mathcal{Z}^m$ is considered $\epsilon$-differentially private if for any subset $\mathcal{R} \subset$ Range$(\mathcal{A})$ and for any datasets $S, S' \in \mathcal{Z}^m$ differing in at most one element (i.e. $\|S - S'\|_1 \leq 1$):

$$\Pr_\phi[h_S^\phi \in \mathcal{R}] \leq e^\epsilon \Pr_\phi[h_{S'}^\phi \in \mathcal{R}], \tag{1}$$

where the probability is taken over the randomness of the algorithm $\mathcal{A}$, with $\phi \sim \Phi$.

**Error Stability** of a possibly randomized algorithm $\mathcal{A}$ for some $\beta > 0$ is defined as Kearns and Ron [1997]:

$$\forall i \in \{1, \cdots, m\}, \quad \left| \mathbb{E}_{\phi,z}[\ell(h_S^\phi, z)] - \mathbb{E}_{\phi,z}[\ell(h_{S \backslash i}^\phi, z)] \right| \leq \beta, \tag{2}$$

where $z \sim \mathcal{D}$ and $\phi \sim \Phi$.

**Generalization.** A randomized algorithm $\mathcal{A}$ is considered to generalize with confidence $\delta$ and at a rate of $\gamma'(m)$ if:

$$\Pr[|R_{emp}(h, S) - R(h)| \leq \gamma'(m)] \geq \delta. \tag{3}$$

**Uniform Model Bias.** The hypothesis $h$ produced by algorithm $\mathcal{A}$ to learn the true conditional $h^* = \mathbb{E}[y|x]$ from a dataset $S \sim \mathcal{D}^m$ has uniform bound on model bias denoted by $\Delta$ if:

$$\forall S \sim \mathcal{D}^m, \quad \left| \mathbb{E}_\phi[R(h_S^\phi) - R(h^*)] \right| \leq \Delta. \tag{4}$$

$\rho$-**Lipschitz Hessian.** The Hessian of $\ell$ is Lipschitz continuous on $\mathcal{Z}$, if $\forall z_1, z_2 \in \mathcal{Z}$, and $\forall h \in$ Range$(\mathcal{A})$, there exists some $\rho > 0$ such that:

$$\|\nabla_{z_1}^2 \ell(h, z_1) - \nabla_{z_2}^2 \ell(h, z_2)\| \leq \rho \|z_1 - z_2\|. \tag{5}$$

**Input Loss Curvature.** As defined by Moosavi-Dezfooli et al. [2019], Garg et al. [2024], input loss curvature is the sum of the eigenvalues of the Hessian $H$ of the loss with respect to input $z_i$. This is conveniently expressed using the trace as:

$$\text{Curv}_\phi(z_i, S) = \text{tr}(H) = \text{tr}(\nabla_{z_i}^2 \ell(h_S^\phi, z_i)) \tag{6}$$

$\upsilon$-**adjacency.** A dataset $S$ is said to contain $\upsilon$-adjacent (read as upsilon-adjacent) elements if it contains two elements $z_i, z_j$ such that $z_j = z_i + \alpha$ for some $\alpha \in B_p(\upsilon)$ (read as $\upsilon$-Ball). This condition can be ensured through construction. Consider a dataset $S'$ which has no $z_j$ s.t $z_j = z_i + \alpha; z_j, z_i \in S'$. We can construct $S$ such that $S = \{z \mid z \in S'\} \cup \{z_i + \alpha\}$ for some $z_i \in S', \alpha \in B_p(\upsilon)$, thereby ensuring $\upsilon$-adjacency.

**Membership Inference Threat Model.** In a membership inference security game [Yeom et al., 2018, Jayaraman et al., 2021, Carlini et al., 2022], a challenger and an adversary interact to test the privacy of a machine learning model. The process begins with the challenger sampling a training dataset from a distribution and training a model on this data. The challenger then flips a coin to decide whether to select a fresh data point from the distribution, which is not part of the training set, or to choose a data point from the training set. This selected data point is given to the adversary, who has access to the same data distribution and the trained model. The adversary's task is to determine whether the given data point was part of the training set or not. If the adversary's guess is correct, the game indicates a successful membership inference attack. Such a game is said to be in a **black-box** setting when the adversary has access to only the challenger's output.

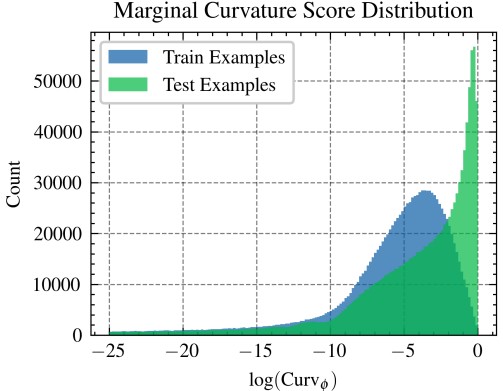
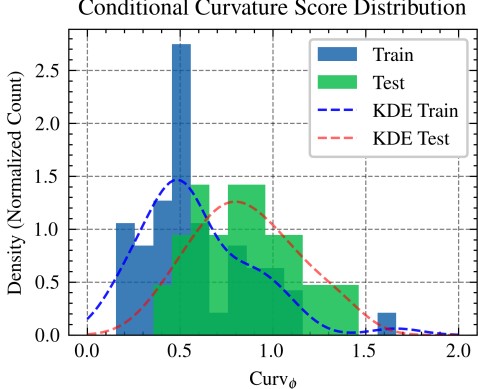

Figure 2: Marginal curvature score histogram for all images in ImageNet when samples in train set vs test set.

Figure 3: Conditional curvature scores for a single image from ImageNet when using 56 shadow models.

## 4 Theoretical Analysis

In this section, we analyze the distinguishability of train-test samples using KL divergence. We do so for two cases, the first case when using network's output probability, and second when using input loss curvature. The importance of analyzing network output probabilities stems from its utilization in the current state-of-the-art attack, LiRA [Carlini et al., 2022]. Thus, studying both cases will let us theoretically analyze and compare their performance with input loss curvature.

Before we begin the analysis, we briefly discuss the conditional and marginal distributions of input loss curvature for train and test examples. Discussing this is important to build intuition for the analysis. Figure 2 visualizes the histogram (proxy for distribution) of input loss curvature for train and test examples from ImageNet [Russakovsky et al., 2015]. Specifically, it plots the log of the input loss curvature $\log(\mathrm{Curv}_\phi)$ obtained on pre-trained ResNet50 [He et al., 2016] models from Feldman and Zhang [2020]. It plots a histogram of these scores, showing the frequency of specific log curvature values.

A naive membership inference attack would apply a threshold to separate these distributions. However, it is common practice to consider sample specific scores (i.e., conditioned on the target sample). This is visualized in Figure 3 which plots the distribution of curvature scores for a single ImageNet sample, indicating differences when the sample is part of the training set versus when it is a test or unseen sample. The data was generated using models from Feldman and Zhang [2020]. The figure also includes a kernel density estimate (KDE, shown as a dashed line) to better to visualize the underlying distribution. Figure 3 suggests that, similar to Carlini et al. [2022] sample conditional curvature scores can be modeled using a Gaussian parametric model. If we represent the curvature score $\mathrm{Curv}_\phi$ as a random variable, then $\mathrm{Curv}_\phi \sim \mathcal{N}(\mu, \sigma)$. The probability density function of $\mathrm{Curv}_\phi$ is a function of the randomness of the algorithm $\phi$, the dataset $S$ and the $i^{th}$ sample $z_i$ and be denoted by $p_c(\phi, S, z_i)$. Similarly, let $p(\phi, S, z_i)$ denote the probability density function of the neural network's output probability, which is also parameterized by the randomness of the algorithm $\phi$, the dataset $S$ and the $i^{th}$ sample $z_i$. With this setup we present the following theoretical results on the upper bound on the KL divergence between train and test distribution for the two cases. Theorem 4.1 presents the upper bound on the KL divergence when using the neural network's output probability scores, and Theorem 4.2 presents the upper bound on the KL divergence when using input loss curvature.

**Theorem 4.1** (Privacy bounds Train-Test KL Divergence). *Assume $\epsilon$-differential private algorithm, then the KL divergence between train-test distributions of the neural network's output probability is upper bound by the differential privacy parameter $\epsilon$ given by:*

$$\mathrm{D_{KL}}\left(p(\boldsymbol{\phi}, S, z_i) \,||\, p(\boldsymbol{\phi}, S^{\setminus i}, z_i)\right) \leq \epsilon \tag{7}$$

**Sketch of Proof.** Given that the algorithm $\mathcal{A}$ is $\epsilon$-differentially private, we know that the probability of the output $h_S^\phi \in \mathcal{R}$ on dataset $S$ is bounded by $e^\epsilon$ times the probability of the same event

on the neighboring dataset $S^{\setminus i}$, i.e. $\mathrm{Pr}_\phi[h_S^\phi \in \mathcal{R}] \leq e^\epsilon \mathrm{Pr}_\phi[h_{S^{\setminus i}}^\phi \in \mathcal{R}]$. From this inequality it follows that the KL divergence between the output distributions of $\mathcal{A}$ on $S$ and $S^{\setminus i}$ is bounded by $\mathrm{KL}(p(\phi, S, z_i) \,||\, p(\phi, S^{\setminus i}, z_i)) \leq \epsilon$. The full proof for Theorem 4.1 is provided in Appendix A.3.

**Theorem 4.2** (Dataset Size and Privacy bound Curvature KL Divergence). *Let the assumptions of error stability 2, generalization 3, and uniform model bias 4 hold. Further, assume $0 \leq \ell \leq L$. Let the conditional distribution be parameterized with variance $\sigma$. Then, any $\epsilon$-differential private algorithm using a dataset of size $m$ with a probability at least $1 - \delta$ satisfies:*

$$\mathrm{D_{KL}}(p_c(\boldsymbol{\phi}, S, z_i) \,||\, p_c(\boldsymbol{\phi}, S^{\setminus i}, z_i)) \leq \frac{[Lm(1 - e^{-\epsilon}) + c]^2}{2\sigma^2} \tag{8}$$

$$c = (4m - 1)\gamma + 2(m - 1)\Delta + \frac{\rho}{6}\mathbb{E}[\|\alpha\|^3] + L \tag{9}$$

**Sketch of Proof.** We begin by assuming a Gaussian model for the curvature distribution and expand the formula for the KL divergence, which leads to expressions involving the mean and standard deviation of the curvature scores. To establish an upper bound for this expression, we prove a lemma concerning the upper bound of the mean curvature, utilizing results from Ravikumar et al. [2024]. This enables us to express the upper bound on the mean curvature in terms of the privacy parameter $\epsilon$. Subsequently, this result is employed to derive an upper bound on the KL divergence of the curvature scores. The proof for Theorem 4.2 is provided in Appendix A.5.

**Discussion.** Theorem 4.1 and Theorem 4.2 provide the upper bound on KL Divergence between train and test distributions of the probability and input loss curvature scores, respectively. They imply the upper limit of MIA performance, and interestingly, the result of Theorem 4.2 suggests the role of the dataset size on MIA performance. However, the relation is not as straightforward as suggested by Equation 8. This is because, in real applications, parameter $\sigma$ depends on $\epsilon$, and so does the loss bound $L$. In fact, the loss bound is also dependent on the number of samples $m$ [Bousquet and Elisseeff, 2002].

**Theorem 4.3** (Dataset Size and Curvature MIA Performance). *Let the assumptions of error stability 2, generalization 3, and uniform model bias 4 hold. Further, assume $0 \leq \ell \leq L$, and the bounds of Theorem 4.1 and 4.2 are tight. Then, the performance of MIA using curvature scores exceeds that of confidence scores with a probability at least $1 - \delta$ when:*

$$m > \frac{(\sqrt{2\sigma^2\epsilon}) - c}{L(1 - e^{-\epsilon})} \tag{10}$$

Theorem 4.3 suggests that the performance of curvature based MIA will exceed that of probability score based methods when the size of the dataset used to train the target model exceed a certain threshold. Indeed, this is what we observe in practice (see section 6.4).

**Sketch of Proof.** The proof compares the upper bounds from Theorem 4.1 and Theorem 4.2, followed by a series of algebraic manipulations. This identifies the conditions under which the performance of curvature scores exceeds that of confidence scores. The full proof can be found in Appendix A.6.

**On the validity of the assumptions.** Before presenting our experiments to validate our theory, we briefly discuss the validity of our assumptions in practical settings. Research by Hardt et al. [2016] shows that stochastic gradient methods, such as stochastic gradient descent, achieve small generalization error and exhibit uniform stability. Thus, the assumptions of stability (Equation 2) and generalization (Equation 3) are justified. Model bias is a property of the model, and a uniform bound across different datasets seems reasonable. Note, uniform loss bound (and $L$ independent of $m$ as used by Theorem 4.3) holds true for certain losses and for statistical models and is often assumed in learning theory [Wang et al., 2016]. Lastly, the $\upsilon$-adjacency can be ensured through construction. For large datasets this may not be needed, because the size of the ball $B_p(\upsilon)$ is unconstrained. Hence two samples from the same class that are similar may suffice. Given the size of modern datasets, this assumption is also reasonable.

# 5 Zero Order Input Loss Curvature MIA

To test if input loss curvature based membership inference performs better than existing methods we need an efficient technique to estimate curvature. We are interested in black-box membership

inference attacks, where one does not have access to the target network's parameters. However, current techniques use Hutchinson's trace estimator [Hutchinson, 1989] to measure input loss curvature such as from Garg and Roy [2023], Garg et al. [2024] or Ravikumar et al. [2024]. This approach needs to evaluate the gradient and hence requires access to model parameters. To solve this issue, we propose using a zero-order estimation technique. Zero-order curvature estimation starts with a finite-difference estimation. Consider a function $f : \mathbb{R}^n \to \mathbb{R}$, then the Hessian at a given point $z_i$ can be estimated as follows:

$$\nabla^2 f(z_i) = n^2 \frac{f(z_i + hv + hu) - f(z_i - hv + hu) - f(z_i + hv - hu) + f(z_i - hv - hu)}{4h^2} uv^\top$$

(11)

where $h$ is a small increment (a hyper parameter in out case), and $u, v$ are vectors in $\mathbb{R}^n$. In our case, to get the input loss curvature, we have $f \leftarrow \ell(g(x_i), y_i)$, where $g$ is the neural network, $\ell$ is the loss function and $z_i = (x_i, y_i)$ are the image, label pair. The pseudo-code for obtaining input loss curvature score using zero order estimation shown in Algorithm 1 in Appendix A.1. To execute membership inference attack using input loss curvature scores, we propose the following methodology. First, we begin by training shadow models, similar to Shokri et al. [2017], Carlini et al. [2022]. These shadow models are used to obtain empirical estimates of parametric model for the curvature score described in Section 4. During the inference phase, we employ a likelihood ratio between the sample being in the train set vs test set parametric models to identify the membership status of a given sample (see Appendix A.2 for pseudo-code). In addition, we perform a negative log likelihood ratio test. We denote the results of likelihood test as 'LR' and the results of the negative log likelihood ratio test as 'NLL'.

# 6 Experiments

## 6.1 Experimental Setup

**Datasets.** To evaluate our theory, we consider the classification task using standard vision datasets. Specifically, we use the CIFAR10, CIFAR100 [Krizhevsky et al., 2009] and ImageNet [Russakovsky et al., 2015] datasets.

**Architectures.** For experiments on ImageNet we use the ResNet50 architecture [He et al., 2016]. For CIFAR10 and CIFAR100, we used the ResNet18 architecture. Details regrading hyperparameters are provided in Appendix A.13. To improve reproducibility, we have open sourced the code at https://github.com/DeepakTatachar/Curvature-Clues.

**Training.** For experiments using private models, we trained ResNet18 models with the Opacus library [Yousefpour et al., 2021] using DP-SGD with a maximum gradient norm of 1.0 and a privacy parameter of $\delta = 1 \times 10^{-5}$. Shadow models for CIFAR10 and CIFAR100 were trained on a 50% subset of the data for 300 epochs. For ImageNet, we used pre-trained models from Feldman and Zhang [2020], trained on a 70% subset of ImageNet. More details about training and compute resources are provided in Appendix A.13.

**Metrics.** To evaluate curvature scores, we use AUROC and balanced accuracy. The Receiver Operating Characteristic (ROC) is the plot of the True Positive Rate (TPR) against the False Positive Rate (FPR). The area under the ROC is called Area Under the Receiver Operating Characteristic (AUROC). AUROC of 1 denotes an ideal detection scheme, since the ideal detection algorithm results in 0 false positive and false negative samples.

## 6.2 Membership Inference

In this subsection, we compare the performance of the proposed input loss curvature based membership inference against prior MIA techniques.

**Setup:** We use CIFAR10, CIFAR100 and ImageNet datasets to test the MIA performance. We consider a **black-box** MIA setup similar to Carlini et al. [2022]. We use ResNet18 for CIFAR10 and CIFAR100, for ImageNet we use the ResNet50 architecture. For all the membership inference attacks, we compute a full ROC curve and report the results. When using shadow models we 64 for CIFAR10 and CIFAR100 and 52 for ImageNet. The AUROC plot for CIFAR100 for various

| Method | ImageNet | | CIFAR100 | | CIFAR10 | |
|---|---|---|---|---|---|---|
| | Bal. Acc. | AUROC | Bal. Acc. | AUROC | Bal. Acc. | AUROC |
| **Curv ZO NLL (Ours)** | **69.16** ± 0.08 | **77.45** ± 0.09 | **84.47** ± 0.21 | **93.49** ± 0.18 | **61.92** ± 0.87 | **68.82** ± 1.30 |
| Curv ZO LR (Ours) | 68.76 ± 0.04 | 72.28 ± 0.04 | 80.48 ± 0.10 | 90.15 ± 0.04 | 55.00 ± 0.17 | 58.89 ± 0.38 |
| Carlini et al. [2022] | 66.14 ± 0.01 | 73.46 ± 0.02 | 81.55 ± 0.13 | 88.89 ± 0.16 | 58.23 ± 0.29 | 61.73 ± 0.32 |
| Yeom et al. [2018] | 58.50 ± 0.02 | 63.23 ± 0.03 | 76.29 ± 0.39 | 82.11 ± 0.31 | 55.57 ± 0.52 | 60.44 ± 0.75 |
| Sablayrolles et al. [2019] | 66.93 ± 0.05 | 76.50 ± 0.04 | 70.22 ± 0.41 | 81.11 ± 0.39 | 56.65 ± 0.56 | 61.50 ± 0.79 |
| Watson et al. [2022] | 61.40 ± 0.06 | 69.44 ± 0.05 | 62.71 ± 0.31 | 71.66 ± 0.50 | 54.86 ± 0.59 | 58.58 ± 0.86 |
| Ye et al. [2022] | 66.16 ± 0.02 | 75.79 ± 0.05 | 80.73 ± 0.24 | 90.88 ± 0.19 | 59.62 ± 0.84 | 67.30 ± 1.25 |
| Song and Mittal [2021] | 57.88 ± 0.03 | 63.29 ± 0.03 | 75.58 ± 0.29 | 82.28 ± 0.27 | 55.63 ± 0.61 | 60.42 ± 0.85 |

Table 1: Comparison of the proposed curvature score based MIA with prior methods tested on ImageNet, CIFAR100, and CIFAR10 datasets. Results reported are the mean ± std obtained over 3 seeds. For CIFAR10 and CIFAR100 64 shadow models were used and 52 for ImagNet.

methods are shown in Figure 4. Table 1 reports the average balanced accuracy and AUROC values over three seeds on various MIA methods on the three datasets. The plot in Figure 4 is a log-log plot to emphasize performance of the proposed method at very low false positive rates (see the orange line y-intercept). We also studied the effect of augmentations and the results can be found in Appendix A.7, the takeaway was that adding more augmentations improved performance. Note that the results presented in Table 1 used 2 augmentations (image + mirror) for our technique and Carlini et al. [2022] for fair comparison.

**Results:** From Table 1, we see that the proposed method performs better than all existing MIA techniques on both ImageNet and CIFAR datasets. Apart from AUROC and balanced accuracy results, the log-log plot emphasizes the performance at really low FPR (i.e. the y intercept, high TPR at low FPR in Figure 4). Additional results at low FPR are available in Appendix A.8.

**Takeaways:** As predicted by Theorem 4.2, the higher KL divergence between train and test curvature score distributions results in superior MIA performance. Further, we observe that while using a negative log likelihood ratio test does better in AUROC and balanced accuracy, the parametric likelihood ratio test does better at low false positive rates as shown in Figure 4. Thus, the proposed use of curvature should be tailored based on use case. Applications that demand high AUROC can use NLL approach, while those that demand high TPR at very low FPR should use the LR technique.

### 6.3 Effect of Privacy

In this section, we study the effect of differential privacy on MIA performance and test the $\epsilon$ relation predicted by Theorem 4.2.

**Setup**: To study the effect of privacy on MIA attack performance, we use DP-SGD trained models. We use privacy $\epsilon$ values of 1, 2, 3, 4, 5, 10, 15, 20, 25, 30, 35, 40, 45, 50 with $\delta = 1 \times 10^{-5}$. We

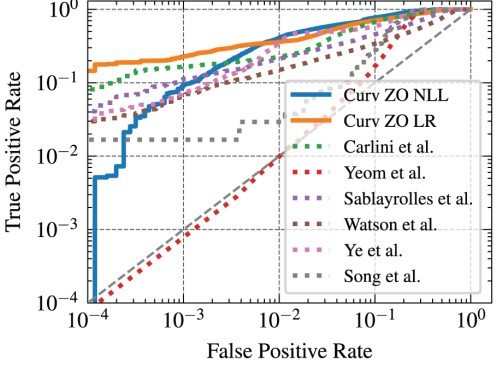

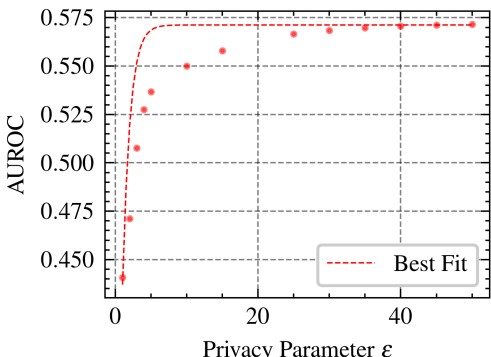

Figure 4: Comparing our method against existing techniques at *low FPR*. The proposed parametric Curv LR technique has the highest TPR at very low FPR.

Figure 5: Validating the upper bound from Theorem 4.2 by fitting the MIA performance (AUROC of Curv LR) on DP-SGD trained models for various privacy parameters $\epsilon$ values.

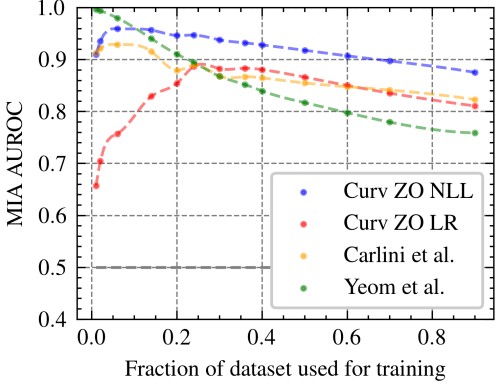
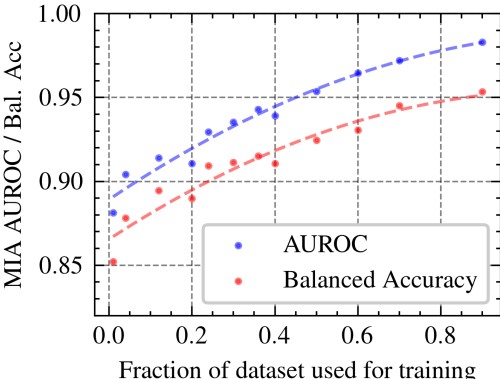

Figure 6: Visualizing MIA performance as a function of the size of the train set, which is randomly sampled.

Figure 7: MIA performance as a function of the size of the train set when subsets contain the lowest curvature examples from the entire set.

plot the result of the proposed curvature based MIA in Figure 5. In Figure 5 we also plots the best fit using $s_f(L_f(1 - e^{-\epsilon}) + c_f)^2$, where $s_f$, $L_f$ and $c_f$ are fit to the data.

**Results and Takeaways:** We see that the theoretical prediction from Theorem 4.2 about MIA performance is well matched. Since Theorem 4.2 provides an upper bound, the results validate the theory.

### 6.4 Effect of Dataset Size

In this section, we study the effect of dataset size on MIA attack performance. This lets us test the relationship of MIA performance to $m$ as predicted by Theorem 4.2 and Theorem 4.3.

**Setup:** For this experiment, we train models on increasing dataset size on CIFAR100. Specifically, we train multiple seeds on various subsets randomly chosen from the CIFAR100 training set. We also repeated the same by choosing the lowest curvature samples from CIFAR100 and train with the same subset sizes. Next, for each of the models we perform MIA attack and we present the results in Figure 6 (randomly chosen) and Figure 7 (lowest curvature samples).

**Results:** From Figure 6, we see that when samples are randomly chosen, the MIA performance decreases as we add more samples. This result is consistent with prior literature [Abadi et al., 2016]. Further as predicted by Theorem 4.3 beyond $30 - 40\%$ subset Curv ZO LR out perform prior works and Curv ZO NLL outperforms prior works beyond $10\%$ subset size.

To extend this analysis, we train models on curvature based coresets [Garg and Roy, 2023]. These coresets of low input loss curvature samples have been shown to memorize less [Garg et al., 2024, Ravikumar et al., 2024]. Thus we expect MIA accuracy to increase as we increase coreset size. This is exactly what happens and is shown in Figure 7 which plots the AUROC and accuracy of the NLL curvature attack as we increase curvature coreset size. However, the MIA performance is higher for comparable size in Figure 6 and 7. **This suggests that curvature based coresets result in more susceptible models**, which is also supported by results in Song et al. [2019].

**Takeaways:** We validate Theorem 4.3. We note that beyond a certain dataset size (of about $30 - 40\%$ subset of the training set Curv ZO LR out perform prior works and Curv ZO NLL outperforms prior works beyond $10\%$ subset size) curvature scores outperform probability score based MIA method. While Theorem 4.3 predicts that curvature-based MIA outperforms other methods on sufficiently large datasets. This condition is often met by real datasets as evident from the results presented.

## 7 Conclusion

In this paper, we explored the properties of input loss curvature on the test set. Specifically, we focus on using input loss curvature to distinguish between train and test examples. We established a theoretical framework deriving an upper bound on train-test KL Divergence based on privacy and

training set size. To extend the applicability of input loss curvature computation to a black-box setting, we propose a novel zero-order curvature estimation method. This enables the development of a new cutting-edge black-box membership inference attack (MIA) methodology that leverages input loss curvature. Through extensive experiments on the ImageNet and CIFAR datasets, we demonstrate that our input loss curvature-based MIA method outperforms existing state-of-the-art techniques. Our results corroborate our theoretical predictions regarding the relationship between MIA performance and dataset size. Specifically, we show that beyond a certain dataset size, the effectiveness of curvature scores surpasses other methods. Importantly, we observe that this dataset size condition is frequently met in practical scenarios, as evidenced by our results on the CIFAR100 dataset. These findings advance our understanding of input loss curvature in the context of privacy and help build more secure deep learning models.

## Acknowledgment

This work was supported in part by, the Center for the Co-Design of Cognitive Systems (COCOSYS), a DARPA-sponsored JUMP 2.0 center, the Semiconductor Research Corporation (SRC), and the National Science Foundation.

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

# A Appendix

## A.1 Zero Order Curvature Estimation

We present the pseudo-code for zero-order curvature score calculation below. Please note that Algorithm 1 shows the input loss curvature calculation for one example; however, it can be easily implemented for batch data.

---
**Algorithm 1** Pseudo-code for Zero Order Input Loss Curvature Estimation
---
1: **Input:** $z_i = (x_i, y_i)$ (image, label data), $\ell$ (loss function), $n_{iter}$ (hyperparameter: number of iterations), $h$ (hyperparameter: small increment), $g$ (neural network)
2: **Output**: $curv\_estimate$ (zero order input loss curvature estimate of $z_i$)
3: $curv \leftarrow 0$
4: **for** $i \leftarrow 1$ **to** $n_{iter}$ **do**
5:     $v \sim$ Rademacher
6:     $u \sim$ Rademacher
7:     $f \leftarrow \ell(g(x_i), y_i)$
8:     $\nabla^2 f(z_i) \leftarrow$ Use Equation 11
9:     $curv \leftarrow curv + \text{tr}(\nabla^2 f(z_i))$
10: **end for**
11: $curv\_estimate \leftarrow curv/n_{iter}$
12: **return** $curv\_estimate$

---

## A.2 Curvature based MIA

In this section, we present the detailed step by step description and pseudo code for the curvature based MIA (refer to Algorithm 2)

1. Initialization (Lines 3-4): Two empty sets, $curv_{in}$ and $curv_{out}$, are initialized. These sets will store the curvature scores for the shadow models trained with and without the target example $z_i$.

2. Shadow Model Training (Lines 5-12): For a specified number of iterations $N$, the algorithm performs the following steps:
   - Sampling Shadow Dataset (Line 6): A shadow dataset $\mathcal{D}_{attack}$ is sampled from the data distribution $\mathcal{D}$.
   - Training IN Shadow Model (Line 7): A shadow model $g_{in}^n$ is trained on the shadow dataset augmented with the target example, $\mathcal{D}_{attack} \cup \{z_i\}$. The curvature score for this model, denoted as $curv_{in}^n$, is computed using Algorithm 1.
   - Collecting IN Curvature Scores (Line 8): The curvature score for the training set $\mathcal{D}_{attack} \cup \{z_i\}$ of this model $curv_{in}^n$ is added to the set $curv_{in}$.
   - Training OUT Shadow Model (Line 9): Another shadow model $g_{out}^n$ is trained on the shadow dataset excluding the target example, $\mathcal{D}_{attack} \setminus \{z_i\}$. The curvature score for this model, denoted as $curv_{out}^n$, is also computed using Algorithm 1.
   - Collecting OUT Curvature Scores (Line 10): The curvature score for the training set $\mathcal{D}_{attack} \setminus \{z_i\}$ of this model $curv_{out}^n$ is added to the set $curv_{out}$.

3. Model Parameter Estimation (Lines 13-16): After training the shadow models, the algorithm calculates the mean ($\mu_{in}$, $\mu_{out}$) and variance ($\sigma_{in}^2$, $\sigma_{out}^2$) of the collected curvature scores for both the 'IN' and 'OUT' models.

4. Target Model Query (Line 17): The curvature score for the target model $g_{target}$ with respect to the target example $z_i$, denoted as $curv_{target}$, is computed using Algorithm 1.

5. Likelihood Ratio Test (Line 18-19): The likelihood ratio $P_{in}$ is calculated. This ratio compares the probability of the observed curvature score under the 'IN' distribution against the 'OUT' distribution and is returned. For the 'NLL' version $P_{in}$ us given by:

$$P_{in}^{NLL} = \log\left[P\left(curv_{target} \mid \mathcal{N}(\mu_{out}, \sigma_{out}^2)\right)\right] - \log\left[P\left(curv_{target} \mid \mathcal{N}(\mu_{in}\sigma_{in}^2)\right)\right] \tag{12}$$

**Algorithm 2** Membership Inference Attack using Input Loss Curvature Scores

---

**Require:** Target model $g_{target}$, target example $z_i = (x_i, y_i)$, data distribution $\mathcal{D}$
1: $curv_{in} \leftarrow \{\}$
2: $curv_{out} \leftarrow \{\}$
3: **for** $n$ in $N$ **do**
4:     $\mathcal{D}_{attack} \leftarrow \mathcal{D}$                                                    ▷ Sample a shadow dataset
5:     $g_{in}^n \leftarrow$ Train on $\mathcal{D}_{attack} \cup \{z_i\}$.                              ▷ Train in shadow model
6:     $curv_{in}^n \leftarrow$ Use Algorithm 1
7:     $curv_{in} \leftarrow curv_{in} \cup curv_{in}^n$
8:     $g_{out}^n \leftarrow$ Train on $\mathcal{D}_{attack} \setminus \{z_i\}$.                        ▷ Train out shadow model
9:     $curv_{out}^n \leftarrow$ Use Algorithm 1
10:     $curv_{out} \leftarrow curv_{out} \cup curv_{out}^n$
11: **end for**
12: $\mu_{in} \leftarrow \text{mean}(curv_{in})$
13: $\mu_{out} \leftarrow \text{mean}(curv_{out})$
14: $\sigma_{in}^2 \leftarrow \text{var}(curv_{in})$
15: $\sigma_{out}^2 \leftarrow \text{var}(curv_{out})$
16: $curv_{target} \leftarrow$ Use Algorithm 1 with $g_{target}$ and $z_i$.
17: $P_{in} = \dfrac{P\left(curv_{target} \mid \mathcal{N}(\mu_{in}, \sigma_{in}^2)\right)}{P\left(curv_{target} \mid \mathcal{N}(\mu_{out}, \sigma_{out}^2)\right)}$
18: **return** $P_{in}$

---

### A.3   Proof of Theorem 4.1

Consider an $\epsilon-$DP algorithm and $S, S^{\setminus i}$ such that $||S - S^{\setminus i}|| = 1$. Next let $\mathcal{R} \subset \text{Range}(\mathcal{A})$ such that $\mathcal{R} = \{h \mid h(x_i) = y_i\}$. Since $\mathcal{A}$ is $\epsilon$-differentially private, then it follows from the definition of differential privacy in Equation 1 that

$$\Pr_\phi[h_S^\phi \in \mathcal{R}] \le e^\epsilon \Pr_\phi[h_{S^{\setminus i}}^\phi \in \mathcal{R}] \tag{13}$$

$$\implies p(\phi, S, z_i) \le e^\epsilon p(\phi, S^{\setminus i}, z_i) \tag{14}$$

Since $p(\phi, S, z_i) = \Pr_\phi[h_S^\phi \in \mathcal{R}]$ and denotes the probability density function of the neural network's output probability. Now, we use the definition of KL divergence

$$D_{\text{KL}}\left(p(\phi, S, z_i) \,||\, p(\phi, S^{\setminus i}, z_i)\right) = \mathbb{E}_\phi\left[\log\left(\frac{p(\phi, S, z_i)}{p(\phi, S^{\setminus i}, z_i)}\right)\right]$$

$$\le \mathbb{E}_\phi\left[\log e^\epsilon\right] \quad \text{From Equation 14}$$

$$D_{\text{KL}}\left(p(\phi, S, z_i) \,||\, p(\phi, S^{\setminus i}, z_i)\right) \le \epsilon \quad \blacksquare$$

### A.4   Lemma A.1

**Lemma A.1** (Upper bound on Input Loss Curvature). *Let $\mathcal{A}$ be a randomized algorithm which is $\epsilon$-differentially private and the assumptions of error stability 2, generalization 3, and uniform model bias 4 hold. Further, assume $0 \le \ell \le L$. Then for two adjacent datasets $S, S^{\setminus i} \sim \mathcal{D}$ with a probability at least $1 - \delta$ we have:*

$$\mathbb{E}_\phi[\text{Curv}_\phi(z_i, S^{\setminus i})] \le L(m(1 - e^{-\epsilon}) + 1) + c_1 \tag{15}$$

$$c_1 = (4m - 1)\gamma + 2(m - 1)\Delta + \frac{\rho}{6}\mathbb{E}[\|\alpha\|^3] \tag{16}$$

**Proof of Lemma A.1** For this proof, we use results from Nesterov and Polyak [2006], we restate it here for convenience

**Lemma A.2.** *If Lipschitz assumption 5 on the Hessian of $\ell$ holds from Nesterov and Polyak [2006] we have*

$$|\ell(h, z_1) - \ell(h, z_2) - \langle \nabla\ell(h, z_2), z_1 - z_2 \rangle - \langle \nabla^2\ell(h, z_2)(z_1 - z_2), z_1 - z_2 \rangle| \le \frac{\rho}{6}|z_1 - z_2|^3 \tag{17}$$

From Lemma A.2 we have

$$-\frac{\rho}{6}|z_1 - z_2|^3 \leq \ell(h, z_1) - \ell(h, z_2) - \langle \nabla \ell(h, z_2), z_1 - z_2 \rangle - \langle \nabla^2 \ell(h, z_2)(z_1 - z_2), z_1 - z_2 \rangle$$

$$\ell(h, z_1) - \ell(h, z_2) - \langle \nabla \ell(h, z_2), z_1 - z_2 \rangle - \langle \nabla^2 \ell(h, z_2)(z_1 - z_2), z_1 - z_2 \rangle \leq \frac{\rho}{6}|z_1 - z_2|^3$$

This gives us a lower bound on $\ell(h, z_1)$

$$-\frac{\rho}{6}|z_1 - z_2|^3 + \ell(h, z_2) + \langle \nabla \ell(h, z_2), z_1 - z_2 \rangle + \langle \nabla^2 \ell(h, z_2)(z_1 - z_2), z_1 - z_2 \rangle \leq \ell(h, z_1)$$
(18)

Consider $z_j \in S$ such that $z_i = z_j + \alpha$ for some $j \neq i$ where $\alpha \in B_p(v)$ such that $\mathbb{E}[\alpha] = 0$ and $\mathbb{E}[\alpha^T \alpha] = 1$. Using the lower bound in Lemma A.2 from Ravikumar et al. [2024] with $z_1 = z_i, z_2 = z_j$ we get

$$\mathbb{E}_\phi[\ell(h_{S\backslash i}^\phi, z_j)] - \mathbb{E}_\phi[\ell(h_S^\phi, z_i)] \leq m\beta + (4m - 1)\gamma + 2(m - 1)\Delta$$
(19)

Using the result from Lemma A.2

$$-\frac{\rho}{6}\|\alpha\|^3 + \mathbb{E}_\phi[\ell(h_{S\backslash i}^\phi, z_i)] + \mathbb{E}_\phi[\langle \nabla \ell(h_{S\backslash i}^\phi, z_i), \alpha \rangle]$$
$$+ \mathbb{E}_\phi[\langle \nabla^2 \ell(h_{S\backslash i}^\phi, z_i)\alpha, \alpha \rangle] - \mathbb{E}_\phi[\ell(h_S^\phi, z_i)] \leq m\beta + (4m - 1)\gamma + 2(m - 1)\Delta \quad (20)$$

Taking Expectation over $\alpha$ and since the mean of $\alpha = 0$ we have

$$\mathbb{E}_\phi[\ell(h_{S\backslash i}^\phi, z_i)] + \mathbb{E}_{\alpha,\phi}[\langle \nabla \ell(h_{S\backslash i}^\phi, z_i), \alpha \rangle] + \mathbb{E}_{\alpha,\phi}[\langle \nabla^2 \ell(h_{S\backslash i}^\phi, z_i)\alpha, \alpha \rangle]$$
$$- \mathbb{E}_\phi[\ell(h_S^\phi, z_i)] \leq m\beta + (4m - 1)\gamma + 2(m - 1)\Delta + \frac{\rho}{6}\|\alpha\|^3$$

Note that we can change the order of expectation using Fubini's theorem

$$\mathbb{E}_\phi[\ell(h_{S\backslash i}^\phi, z_i)] + \mathbb{E}_{\phi,\alpha}[\langle \nabla^2 \ell(h_{S\backslash i}^\phi, z_i)\alpha, \alpha \rangle] - \mathbb{E}_\phi[\ell(h_S^\phi, z_i)] \leq m\beta + (4m - 1)\gamma + 2(m - 1)\Delta + \frac{\rho}{6}\|\alpha\|^3$$

$$\mathbb{E}_\phi[\ell(h_{S\backslash i}^\phi, z_i)] + \mathbb{E}_\phi[\mathrm{tr}(\nabla^2 \ell(h_{S\backslash i}^\phi, z_i))] - \mathbb{E}_\phi[\ell(h_S^\phi, z_i)] \leq m\beta + (4m - 1)\gamma + 2(m - 1)\Delta + \frac{\rho}{6}\|\alpha\|^3$$

$$\mathbb{E}_\phi[\mathrm{tr}(\nabla^2 \ell(h_{S\backslash i}^\phi, z_i))] \leq m\beta + (4m - 1)\gamma + 2(m - 1)\Delta + \frac{\rho}{6}\mathbb{E}[\|\alpha\|^3] + \mathbb{E}_\phi[\ell(h_S^\phi, z_i)] - \mathbb{E}_\phi[\ell(h_{S\backslash i}^\phi, z_j)]$$

$$\mathbb{E}_\phi[\mathrm{tr}(\nabla^2 \ell(h_S^\phi, z_i))] \leq m\beta + (4m - 1)\gamma + 2(m - 1)\Delta + \frac{\rho}{6}\mathbb{E}[\|\alpha\|^3] + L$$

From Ravikumar et al. [2024] Lemma 5.2 we have $\beta \leq L(1 - e^{-\epsilon})$. Thus we have the upper bound:

$$\mathbb{E}_\phi[\mathrm{Curv}_\phi(z, S^{\backslash i})] \leq L(m(1 - e^{-\epsilon}) + 1) + c_1$$
(21)

$$c_1 = (4m - 1)\gamma + 2(m - 1)\Delta + \frac{\rho}{6}\mathbb{E}[\|\alpha\|^3] \quad \blacksquare$$
(22)

### A.5  Proof of Theorem 4.2

This proof uses results from Lemma A.1 provided above. For the proof of Theorem 4.2 let $p_c(\phi, S, z_i)$ be the curvature probability mass function, then using the Gaussian model discussed in Section 4 we model it as a Gaussian. Thus, we can write

$$p_c(\phi, S, z_i) = \frac{1}{\sigma_S \sqrt{2\pi}} e^{-\frac{(\phi - \mu_S)^2}{2\sigma_S^2}}$$
(23)

$$p_c(\phi, S^{\backslash i}, z_i) = \frac{1}{\sigma_{S\backslash i} \sqrt{2\pi}} e^{-\frac{(\phi - \mu_{S\backslash i})^2}{2\sigma_{S\backslash i}^2}}$$
(24)

Now consider the $D_{KL}(p_c(\phi, S, z_i) \,||\, p_c(\phi, S^{\setminus i}, z_i))$

$$D_{KL}(p_c(\phi, S, z_i) \,||\, p_c(\phi, S^{\setminus i}, z_i)) = \mathbb{E}_\phi \left[ \log \left( \frac{p_c(\phi, S, z_i)}{p_c(\phi, S^{\setminus i}, z_i))} \right) \right] \tag{25}$$

$$= \mathbb{E}_\phi \left[ -\frac{(\phi - \mu_S)^2}{2\sigma^2} + \frac{(\phi - \mu_{S^{\setminus i}})^2}{2\sigma^2} \right] \quad \text{Assume } \sigma = \sigma_{S^{\setminus i}} = \sigma_S \tag{26}$$

$$= \mathbb{E}_\phi \left[ \frac{-\phi^2 - \mu_S^2 + 2\phi\mu_S + \phi^2 + \mu_{S^{\setminus i}}^2 - 2\phi\mu_{S^{\setminus i}}}{2\sigma^2} \right] \tag{27}$$

$$= \mathbb{E}_\phi \left[ \frac{\mu_{S^{\setminus i}}^2 - \mu_S^2}{2\sigma^2} \right] \tag{28}$$

$$= \frac{\mu_{S^{\setminus i}}^2 - \mu_S^2}{2\sigma^2} \tag{29}$$

To upper bound Equation 29, we need an upper bound on $\mu_{S^{\setminus i}}$ and a lower bound on $\mu_S$. The lower bound on $\mu_S^2 = 0$

For ease of notation let's define $\beta_{max} := L(m(1 - e^{-\epsilon}) + 1)$. Notice that the upper bound of per sample $\mu_{S^{\setminus i}} = \mathbb{E}_\phi[\text{Curv}_\phi(z, S^{\setminus i})]$. Using the result from Lemma A.1 in Equation 29 we have

$$D_{KL}(p_c(\phi, S, z_i) \,||\, p_c(\phi, S^{\setminus i}, z_i)) \leq \frac{(\beta_{max} + c_1)^2 - 0^2}{2\sigma^2} \tag{30}$$

$$\leq \frac{(\beta_{max} + c_1)^2}{2\sigma^2} \tag{31}$$

$$\leq \frac{[L(m(1 - e^{-\epsilon}) + 1) + c_1]^2}{2\sigma^2} \quad \blacksquare \tag{32}$$

## A.6 Proof of Theorem 4.3

Assuming the bounds of Theorem 4.1 and Theorem 4.2 are tight, then using curvature score will outperform neural network probability scores if:

$$D_{KL}(p_c(\phi, S, z_i) \,||\, p_c(\phi, S^{\setminus i}, z_i)) > D_{KL}(p(\phi, S, z_i) \,||\, p(\phi, S^{\setminus i}, z_i)) \tag{33}$$

$$\frac{[L(m(1 - e^{-\epsilon}) + 1) + c_1]^2}{2\sigma^2} > \epsilon$$

$$\frac{[Lm(1 - e^{-\epsilon}) + c]^2}{2\sigma^2} > \epsilon \quad ; c = c_1 + L$$

$$L^2(1 - e^{-\epsilon})^2 m^2 + 2Lc(1 - e^{-\epsilon})m + c^2 - 2\sigma^2\epsilon > 0$$

Using the quadratic formula we have the roots given by:

$$= \frac{-2Lc(1 - e^{-\epsilon}) \pm \sqrt{4L^2c^2(1 - e^{-\epsilon})^2 - 4L^2(1 - e^{-\epsilon})^2(c^2 - 2\sigma^2\epsilon)}}{2L^2(1 - e^{-\epsilon})^2}$$

$$= \frac{-2Lc \pm \sqrt{4L^2c^2 - 4L^2(c^2 - 2\sigma^2\epsilon)}}{2L^2(1 - e^{-\epsilon})}$$

$$= \frac{-Lc \pm \sqrt{L^2c^2 - L^2(c^2 - 2\sigma^2\epsilon)}}{L^2(1 - e^{-\epsilon})}$$

$$= \frac{-c \pm \sqrt{c^2 - (c^2 - 2\sigma^2\epsilon)}}{L(1 - e^{-\epsilon})}$$

$$= \frac{-c \pm \sqrt{2\sigma^2\epsilon}}{L(1 - e^{-\epsilon})}$$

Assuming both roots are real, the larger root is $\frac{\sqrt{2\sigma^2\epsilon}-c}{L(1-e^{-\epsilon})}$ Thus when $m > \frac{\sqrt{2\sigma^2\epsilon}-c}{L(1-e^{-\epsilon})}$ Equation 33 is satisfied. ∎

## A.7 Input Augmentations Ablation

In this section we present the results of using 1 augmentation (only the image) vs 2 augmentations (image + mirror) in Tables 2, 3 and 4 for ImageNet, CIFAR100 and CIFAR10 respectively. For these results, we used 64 shadow models for CIFAR10 and CIFAR100 and 52 shadow models for ImageNet.

| Method | ImageNet | | | |
| | 1 Augmentation | | 2 Augmentations | |
| | Bal. Acc. | AUROC | Bal. Acc. | AUROC |
|---|---|---|---|---|
| Curv ZO NLL (Ours) | $68.09 \pm 0.07$ | $75.84 \pm 0.08$ | $69.16 \pm 0.08$ | $77.45 \pm 0.09$ |
| Curv ZO LR (Ours) | $67.59 \pm 0.05$ | $70.62 \pm 0.04$ | $68.76 \pm 0.04$ | $72.28 \pm 0.04$ |
| Carlini et al. [2022] | $65.11 \pm 0.03$ | $71.96 \pm 0.04$ | $66.14 \pm 0.01$ | $73.46 \pm 0.02$ |

Table 2: Comparison of the proposed curvature score-based MIA with prior methods on ImageNet dataset with 1 and 2 augmentations. Results reported are the mean ± std obtained over 3 seeds.

| Method | CIFAR100 | | | |
| | 1 Augmentation | | 2 Augmentations | |
| | Bal. Acc. | AUROC | Bal. Acc. | AUROC |
|---|---|---|---|---|
| Curv ZO NLL (Ours) | $83.97 \pm 0.23$ | $92.72 \pm 0.23$ | $84.47 \pm 0.21$ | $93.49 \pm 0.18$ |
| Curv ZO LR (Ours) | $77.88 \pm 0.21$ | $88.10 \pm 0.12$ | $80.48 \pm 0.10$ | $90.15 \pm 0.04$ |
| Carlini et al. [2022] | $80.73 \pm 0.16$ | $87.69 \pm 0.18$ | $81.55 \pm 0.13$ | $88.89 \pm 0.16$ |

Table 3: Comparison of the proposed curvature score-based MIA with prior methods on CIFAR100 dataset with 1 and 2 augmentations. Results reported are the mean ± std obtained over 3 seeds.

| Method | CIFAR10 | | | |
| | 1 Augmentation | | 2 Augmentations | |
| | Bal. Acc. | AUROC | Bal. Acc. | AUROC |
|---|---|---|---|---|
| Curv ZO NLL (Ours) | $61.03 \pm 0.84$ | $67.56 \pm 1.23$ | $61.92 \pm 0.87$ | $68.82 \pm 1.30$ |
| Curv ZO LR (Ours) | $54.54 \pm 0.19$ | $58.06 \pm 0.41$ | $55.00 \pm 0.17$ | $58.89 \pm 0.38$ |
| Carlini et al. [2022] | $56.96 \pm 0.27$ | $60.05 \pm 0.34$ | $58.23 \pm 0.29$ | $61.73 \pm 0.32$ |

Table 4: Comparison of the proposed curvature score-based MIA with prior methods on CIFAR10 dataset with 1 and 2 augmentations. Results reported are the mean ± std obtained over 3 seeds.

## A.8 Low FPR Results

In this section, we present the performance of curvature based MIA compared with prior works using TPR (true positive rate) at very low FPR percentages. The results are presented when using 1 augmentation (only the image) and 2 augmentations (image + mirror) in Table 5 and Figure 8. For these results, we use 64 shadow models for all the results in Table 5 (for methods that use shadow models). The results for Carlini et al. [2022] are a little lower than the one reported by Carlini et al. [2022] in Table I of their paper. However, note that the authors of Carlini et al. [2022] report the results with 256 shadow models while we report it using 64 shadow models. This highlights the benefits of curvature based approach, it achieves similar performance to Carlini et al. [2022] with 256 shadow models but with only 64 shadow models thus needs significantly less shadow models to achieve similar performance.

**Takeaways:** The parametric curvature LR method has the best TPR at very low FPR and makes efficient use of shadow models.

| Method | 1 Augmentation | | 2 Augmentations | |
|---|---|---|---|---|
| | TPR @ 0.1% FPR | TPR @ 0.01% FPR | TPR @ 0.1% FPR | TPR @ 0.01% FPR |
| **Curv ZO LR (Ours)** | **21.07** $\pm$ **0.80** | **15.74** $\pm$ **3.12** | **23.92** $\pm$ **0.92** | **17.17** $\pm$ **1.87** |
| Curv ZO NLL (Ours) | 6.42 $\pm$ 0.62 | 0.05 $\pm$ 0.04 | 8.29 $\pm$ 1.52 | 0.10 $\pm$ 0.14 |
| Carlini et al. [2022] | 15.52 $\pm$ 0.54 | 5.02 $\pm$ 1.70 | 15.80 $\pm$ 0.49 | 6.56 $\pm$ 1.13 |
| Sablayrolles et al. [2019] | 10.50 $\pm$ 0.59 | 4.30 $\pm$ 0.53 | 10.50 $\pm$ 0.59 | 4.30 $\pm$ 0.53 |
| Watson et al. [2022] | 6.62 $\pm$ 0.23 | 2.95 $\pm$ 0.18 | 6.62 $\pm$ 0.23 | 2.95 $\pm$ 0.18 |
| Ye et al. [2022] | 6.63 $\pm$ 0.52 | 2.86 $\pm$ 0.42 | 6.63 $\pm$ 0.52 | 2.86 $\pm$ 0.42 |
| Song and Mittal [2021] | 1.21 $\pm$ 0.37 | 1.21 $\pm$ 0.37 | 1.21 $\pm$ 0.37 | 1.21 $\pm$ 0.37 |
| Yeom et al. [2018] | 0.07 $\pm$ 0.01 | 0.01 $\pm$ 0.00 | 0.07 $\pm$ 0.01 | 0.01 $\pm$ 0.00 |

Table 5: Comparison of TPR @ 0.1% FPR and TPR @ 0.01% FPR for CIFAR100 dataset with 1 and 2 augmentations. Results reported are the mean $\pm$ std obtained over 3 seeds and using 64 shadow models.

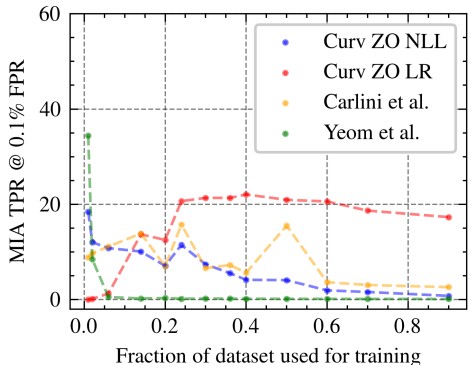

Figure 8: TPR at low FPR visualizing MIA performance as a function of the size of the train set, which is randomly sampled.

Figure 9: Low TPR performance comparison on CIFAR100 of ZO estimation (Curv ZO, black-box) vs. Hutchinson (Curv, white-box).

### A.9 ZO Estimation Performance

In this section, we present the results of comparing the proposed zero-order curvature estimation with Hutchinson's trace estimator-based curvature calculation. It is important to note that the trace estimator proposed by Garg et al. [2024] requires access to the model's parameters, thus it is a white-box attack. In contrast, the proposed zero-order (ZO) estimation operates as a black-box attack.

The comparative results are presented in Table 6. Additionally, Figure 9 illustrates the True Positive Rate (TPR) vs. False Positive Rate (FPR) for the ZO estimation and Hutchinson's trace estimator, utilizing a log-log plot to emphasize performance at low FPR.

**Takeaways.** From Table 6 we see that zero-order approach to curvature estimation results in notable differences in performance. Specifically, when examining AUROC and balanced accuracy, particularly on the CIFAR-10 dataset, a gap of several percentage points can be observed between the two methods. From Figure 9, we observe that the proposed LR technique performs similarly between Hutchinson's trace estimator [Garg et al., 2024] and the proposed ZO estimation. However, the NLL method exhibits much greater sensitivity to estimation errors, as demonstrated by a significant performance drop at low FPR levels.

| Dataset | Method | AUROC | Balanced Accuracy |
|---------|--------|-------|-------------------|
| CIFAR100 | Curv ZO NLL | $93.49 \pm 0.18$ | $84.47 \pm 0.21$ |
| | Curv NLL | $95.01 \pm 0.19$ | $85.66 \pm 0.30$ |
| | Curv ZO LR | $90.15 \pm 0.04$ | $80.48 \pm 0.10$ |
| | Curv LR | $92.87 \pm 0.02$ | $83.75 \pm 0.16$ |
| CIFAR10 | Curv ZO NLL | $68.82 \pm 1.30$ | $61.92 \pm 0.87$ |
| | Curv NLL | $74.20 \pm 1.42$ | $64.96 \pm 1.05$ |
| | Curv ZO LR | $58.89 \pm 0.38$ | $55.00 \pm 0.17$ |
| | Curv LR | $68.53 \pm 0.27$ | $61.82 \pm 0.11$ |

Table 6: Performance Metrics for CIFAR100 and CIFAR10 of zero order estimation (Curv ZO, black-box) vs. Hutchinson trace estimation (Curv, white-box) [Garg et al., 2024].

## A.10 Performance Across Architectures

In this section, we present the results of evaluating the proposed zero-order MIA on different neural network architectures. We selected four architectures: Inception [Szegedy et al., 2015] (specifically, the smaller Inception architecture used by Feldman and Zhang [2020]), VGG11 [Simonyan and Zisserman, 2014], VGG11 with batch normalization (denoted as VGG11BN), and finally, a ViT-B16 vision transformer [Vaswani et al., 2017, Dosovitskiy et al., 2021]. The results are provided in Table Table 7. Please note that for all of the results we used ResNet18 shadow models and 2 augmentations.

**Takeaways.** The proposed techniques outperform the previous state-of-the-art method by Carlini et al. [2022] across all architectures. Additionally, the lower performance on the Vision Transformer (ViT) model can be attributed to transfer learning. Since ViTs do not perform well when trained from scratch on CIFAR-100, we used an ImageNet-pretrained model, which was fine-tuned on CIFAR-100. This led to significant privacy benefits, which aligns with the findings of Mehta et al. [2022].

| Arch. | Method | AUROC | Bal. Acc. | TPR @ 1% FPR |
|-------|--------|-------|-----------|--------------|
| Inception | Curv ZO NLL | $87.77 \pm 0.61$ | $79.04 \pm 0.65$ | $15.33 \pm 1.38$ |
| | Curv ZO LR | $81.94 \pm 0.79$ | $73.46 \pm 0.88$ | $19.46 \pm 0.81$ |
| | Carlini et al. [2022] | $83.77 \pm 0.63$ | $76.70 \pm 0.66$ | $16.66 \pm 0.82$ |
| VGG11 | Curv ZO NLL | $73.68 \pm 2.99$ | $66.53 \pm 3.35$ | $6.19 \pm 1.15$ |
| | Curv ZO LR | $68.29 \pm 1.38$ | $64.30 \pm 1.13$ | $8.14 \pm 1.85$ |
| | Carlini et al. [2022] | $62.85 \pm 0.71$ | $61.08 \pm 0.87$ | $2.07 \pm 0.19$ |
| VGG11BN | Curv ZO NLL | $87.76 \pm 0.17$ | $79.12 \pm 0.11$ | $15.56 \pm 1.34$ |
| | Curv ZO LR | $76.17 \pm 0.65$ | $70.98 \pm 0.26$ | $22.80 \pm 0.40$ |
| | Carlini et al. [2022] | $70.57 \pm 0.39$ | $67.13 \pm 0.43$ | $3.22 \pm 0.34$ |
| ViT-B16 | Curv ZO NLL | $58.10 \pm 0.20$ | $55.40 \pm 0.18$ | $1.60 \pm 0.01$ |
| | Curv ZO LR | $53.98 \pm 0.30$ | $52.55 \pm 0.18$ | $1.91 \pm 0.25$ |
| | Carlini et al. [2022] | $54.59 \pm 0.05$ | $53.22 \pm 0.10$ | $1.44 \pm 0.14$ |

Table 7: MIA performance metrics for different architectures and methods on CIFAR100 dataset.

### A.11 Broader Impact

The development and analysis of input loss curvature in deep neural networks presented in this work have significant implications for both the academic and practical fields of machine learning and privacy. By introducing a novel black-box membership inference attack that leverages input loss curvature, this research advances the state-of-the-art in privacy testing for machine learning models.

From a societal perspective, the ability to detect and mitigate membership inference attacks is crucial for maintaining user privacy in machine learning applications. This work helps pave the way for more robust privacy-preserving techniques, ensuring that sensitive information is protected against unauthorized inference. Furthermore, the insights gained from the relationship between input loss curvature and memorization can guide the development of more secure machine learning models, which is particularly important as these models are increasingly deployed in sensitive domains such as healthcare, finance, and personal data processing.

Additionally, this research highlights the potential of using subsets of training data as a defense mechanism against membership inference attacks. By identifying that training on certain sized subsets can improve resistance to these attacks, this work offers practical guidance for model training practices that can enhance privacy without significantly compromising performance.

### A.12 Limitations

In this paper we presented the a theoretical analysis and experimental evidence for improved MIA using input loss curvature. The theoretical and empirical evidence also shows that certain sized subsets of the training set may provide defense against membership inference attacks. However, as mentioned in the paper, the MIA performance improvement occurs only above a certain training dataset size below which the non-parametric model of 'Curv NLL' works better and is not explained by the theoretical analysis. Similar to techniques that use shadow models such as by Shokri et al. [2017], Carlini et al. [2022] we need to train shadow models, which can be computationally expensive. Further, the method described requires more queries at minimum $4\times$ more than Carlini et al. [2022]. We believe these limitations can be addressed by follow up research.

### A.13 Reproducibility Details

In this section we present additional details for reproducing our results. Additionally we have released our code here `https://github.com/DeepakTatachar/Curvature-Clues`.

**Training.** For experiments that use private models, we use the Opacus library [Yousefpour et al., 2021] to train ResNet18 models for 20 epochs till the privacy budget is reached. We use DP-SGD [Abadi et al., 2016] with the maximum gradient norm set to 1.0 and privacy parameter $\delta = 1 \times 10^{-5}$. The initial learning rate was set to 0.001. The learning rate is decreased by 10 at epochs 12 and 16 with a batch size of 128.

For shadow model training on CIFAR10 and CIFAR100 we trained on 50% randomly sampled subset of the data for 300 epochs with a batch size of 512 for CIFAR100 and 256 for CIFAR10. We used SGD optimizer with the initial learning rate set to 0.1, weight decay of $1 \times 10^{-4}$ and momenutm of 0.9. The learning rate was decayed by 0.1 at $180^{th}$ and $240^{th}$ epoch. For ImageNet we used pre-trained models from Feldman and Zhang [2020] as shadow models which were trained on a 70% subset of ImageNet. For both CIFAR10 and CIFAR100 datasets, we used the following sequence of data augmentations for training: resize $(32 \times 32)$, random crop, and random horizontal flip, this is followed by normalization.

**Testing.** During testing we used resize followed by normalization. We used two augmentations the original image and its mirror. The number of augmentations used are specified in the corresponding experiment section. When using pre-trained models from Feldman and Zhang [2020] we validated the accuracy of the models before performing experiments.

**Compute Resources.** All of the experiments were performed on a heterogeneous compute cluster consisting of 9 1080Ti's, 6 2080Ti's and 4 A40 NVIDIA GPUs, with a total of 100 CPU cores and a combined 1.2 TB of main system memory. However, the results can be replicated with a single GPU with 11GB of VRAM.

**Hyperparameters.** Our code uses 2 hyper parameters for zero-order input loss curvature estimation. The $n_{iter}$ and $h$ in Algorithm 1. We used $n_{iter} = 10$ and $h = 0.001$. To improve reproducibility, we have provided the code in the supplementary material.

## A.14 Licenses for Assets Used

For each of the assets used we present the licenses below we also have provided the correct citation in the main paper as well as here for convenience.

1. ImageNet [Russakovsky et al., 2015]: Terms of access available at `https://image-net.org/download.php`

2. CIFAR10 [Krizhevsky et al., 2009]: Unknown / No license provided

3. CIFAR100 [Krizhevsky et al., 2009]: Unknown / No license provided

4. Pre-trained ImageNet models and code: We used pre-trained ImageNet models from Feldman and Zhang [2020] which is licensed under Apache 2.0 `https://github.com/google-research/heldout-influence-estimation/blob/master/LICENSE`.

5. Baseline methods: We re-implemented the baseline methods hence is provided with along with our code which is distributed under the MIT License.

6. Opacus [Yousefpour et al., 2021]: Licensed under Apache 2.0, `https://github.com/pytorch/opacus/blob/main/LICENSE`.

7. Pytorch [Ansel et al., 2024]: Custom BSD-style license available at `https://github.com/pytorch/pytorch/blob/main/LICENSE`.

8. ResNet Model Architecture [He et al., 2016]: MIT license available at `https://github.com/kuangliu/pytorch-cifar/blob/master/LICENSE`

