# OpenReview forum: "Curvature Clues: Decoding Deep Learning Privacy with Input Loss Curvature"
_NeurIPS.cc/2024/Conference — NeurIPS 2024 spotlight_

### Official Review · Reviewer_2dr3 · 2024-06-30

**Soundness:** 3
**Presentation:** 3
**Contribution:** 3
**Rating:** 6
**Confidence:** 3

**Summary:**

This paper conducts a theoretical and empirical investigation of the use of input loss curvatures in MIAs. They study the train-test input loss curvature scores and use this to develop a zero order input loss curvature black-box MIA.

**Strengths:**

1. This paper provides an interesting theoretical explanation of MIAs based on input loss curvature that have performed well empirically but lacks theoretical understanding.
2. The new MIA is simple and motivated by theory, and the evaluation of it is thorough and clearly shows that this method is more performant that existing baselines.
3. The paper is well-written and clear.

**Weaknesses:**

1. Missing some references, e.g. Li et al. 2023 or the original DetectGPT paper (Mitchell et al. 2023).
2. I believe the paper would benefit from including some sort of proof sketch of the theorems in the main body.

**Questions:**

1. Other authors have used the Hutchinson trace estimator to estimate information about input loss curvature. How does the performance of this method compare to the zero-order method presented in the paper?

**Limitations:**

Yes

---

> ### Author Rebuttal · Authors · 2024-08-06
>
> We thank the reviewer for their feedback, we address the weaknesses and questions below:
>
> 1.	Missing some references, e.g. Li et al. 2023 or the original DetectGPT paper (Mitchell et al. 2023).
>
> A: We thank the reviewer for the feedback, we will include these references in the revised version of the paper.
>
> 2.	I believe the paper would benefit from including some sort of proof sketch of the theorems in the main body.
>
> A: We thank the reviewer for the feedback, we will include a brief sketch of proof in the main body of the paper in the revised version of the paper.
>
> 3.	Other authors have used the Hutchinson trace estimator to estimate information about input loss curvature. How does the performance of this method compare to the zero-order method presented in the paper?
>
> A: We provide additional results on the effect of zero-order estimation. We run the proposed MIA attack on CIFAR100 and CIFAR10 datasets with zero order (ZO) estimation (black-box) and compare it against white-box Hutchinson curvature calculation [Garg et al.]. The results are provided in Table 1 and Figure 3 in the rebuttal pdf (please see the global response). Please note this comparison is a black-box (ours) vs white-box (Hutchinson) comparison hence is provided for a complete analysis.

---

> > ### Comment · Reviewer_2dr3 · 2024-08-09
> >
> > I have read the rebuttal and will increase my score.

---

### Official Review · Reviewer_1uCf · 2024-07-12

**Soundness:** 3
**Presentation:** 3
**Contribution:** 3
**Rating:** 7
**Confidence:** 3

**Summary:**

This paper builds the connection between privacy and input loss curvature.  Core observation is that test samples often lie in higher curvature areas than those prototypical samples, so the input loss curvature can be a good metric to distinguish between train and test sets. Utilizing this observation, authors build a stronger black-box MIA, outperforming previous MIAs, and they also reveal a previously unknown limitation of shadow model based MIAs. The paper provides both theoretical analysis and experimental results.

**Strengths:**

S1. The introduction of input loss curvature into the area of privacy attacks are original and novel. Previous works have revealed the utility of input loss curvature on coreset selection, adversarial robustness and model memorization, this work has further broadened its application.

S2. This work proposed an advanced black-box MIA, which is more effective than previous attacks. Moreover, the study reveals one limitation of shadow-model based MIAs: when the dataset size used to train shadow models is similar to dataset size used to train target model, shadow-model based MIAs perform poorly.

**Weaknesses:**

Line 117 missing a `.`

**Questions:**

Q1: What is $s_f$, $L_f$ and $c_f$ in Line 300? The notation seems to be missing from the context.

Q2: How accurate is the zeroth-order estimation of input loss curvature? Can you provide a comparison of this estimation to the real values   (assuming we have white-box access to target model)

**Limitations:**

The limitations of this work is properly discussed in Appendix.

---

> ### Author Rebuttal · Authors · 2024-08-06
>
> We thank the reviewer for their feedback, we address the weaknesses and questions below:
>
> 1.	What is $s_f$, $L_f$ and $c_f$ in Line 300? The notation seems to be missing from the context. Line 117 missing a .
>
> A: $s_f$, $L_f$ and $c_f$ are the fitting coefficients for equation 8, we will clarify this in the revised version of the paper and include missing punctuation in line 117.
>
> 2. How accurate is the zeroth-order estimation of input loss curvature? Can you provide a comparison of this estimation to the real values (assuming we have white-box access to target model)
>
> A: We provide additional results on the effect of zero-order estimation. We run the proposed MIA attack on CIFAR100 and CIFAR10 datasets with zero order (ZO) estimation (black-box) and compare it against white-box Hutchinson curvature calculation [Garg et al.]. The results are provided in Table 1 and Figure 3 in the rebuttal pdf (see global response). Please note this comparison is a black-box (ours) vs white-box (Garg et al) comparison hence is provided for a complete analysis. Additionally, please see our global response.

---

> > ### Comment · Reviewer_1uCf · 2024-08-11
> >
> > Thanks for the rebuttal. I will keep my score.

---

### Official Review · Reviewer_mnxr · 2024-07-12

**Soundness:** 2
**Presentation:** 3
**Contribution:** 3
**Rating:** 6
**Confidence:** 4

**Summary:**

This paper develop a theoretical framework to derive an upper bound on train-test distinguishability based on input loss curvature. The authors propose a new black-box membership inference attack utilizing input loss curvature, which surpasses existing methods in membership inference effectiveness. The paper also explores the potential of using subsets of training data as a defense mechanism against shadow model-based membership inference attacks.

**Strengths:**

Membership inference attack is an important and interesting problem and this paper discussed it from a theoretical perspective. The attacking method based on theoretical guarantees makes the method more solid. In the experiment, the curvature based attack proposed by this paper also exceeds other existing methods.

**Weaknesses:**

1. In the theoretical analysis, the authors propose the train-test distinguishability distribution upper bound for both data and curvature. However, they compare the upper bounds directly and suggest that curvature based MIA is a better method when the size of dataset is large enough. The comparison between upper bounds may not reflect the real relationship. To validate this result, one may compare between an upper bound and a lower bound.
2. The assumption that the curvature score can be modeled with Gaussian distribution has not been discussed and validated in the paper, which is a very important assumtion in theoretical analysis. Figure 3 seems that the distribution is not very similar to Gaussian.

**Questions:**

What is the effect of the error in the zeroth-order curvature estimation? How the estimated curvature deviates Gaussian and can we still model the distribution with it?

**Limitations:**

The authors have addressed the limitations of their work by acknowledging that their analysis only holds when the dataset size is large enough. They emphasize the need for further exploration of the method's applicability to different models and datasets and the potential impact of computational complexity.

---

> ### Author Rebuttal · Authors · 2024-08-06
>
> We thank the reviewer for their feedback, we address the weaknesses and questions below:
>
> 1.	In the theoretical analysis, the authors propose the train-test distinguishability distribution upper bound for both data and curvature. However, they compare the upper bounds directly and suggest that curvature based MIA is a better method when the size of dataset is large enough. The comparison between upper bounds may not reflect the real relationship. To validate this result, one may compare between an upper bound and a lower bound.
>
> A: We agree with the reviewer that comparing the upper bound with a lower bound would be a more elegant approach. However, to account for the fact the we compare upper bounds, we provide empirical evidence to validate this result in Figure 6 (please see the updated plot in the rebuttal pdf, specifically Figures 1 and 2). Additionally, please see the global response bullet 2 and the pdf with revised dataset size vs performance plots validating the theoretical result. In summary, we use empirical results to bolster and check the validity of the theoretical result. The results show we perform better than other methods and as predicted by theory.
>
> 2. How the estimated curvature deviates Gaussian and can we still model the distribution with it?
>
> A: The validity of the Gaussian assumption is difficult to assess using a single plot, such as Figure 3 in the main paper, which depicts one image from ImageNet. It was provided as a visual aid for the reader. Parametric MIA methods demonstrate the 'goodness' of parametric estimation in the low FPR region, as shown by Carlini et al. Specifically, accurate parametric estimation results in good low FPR performance [Carlini et al]. Our results in Figure 4 and Table 1 empirically validate that the Gaussian assumption is a good parametric estimate. That is we see the Curv LR method which is the parametric model outperforms the Curv NLL method and others in low FPR performance.
>
> 3. What is the effect of the error in the zeroth-order curvature estimation?
>
> A: We provide additional results on the effect of zero-order estimation. We run the proposed MIA attack on CIFAR100 and CIFAR10 datasets with zero order (ZO) estimation (black-box) and compare it against white-box Hutchinson curvature calculation [Garg et al.]. The results are provided in Table 1 and Figure 3 in the rebuttal pdf. Please note this comparison is a black-box and white-box comparison hence is provided to illustrate the effect of the error in the zeroth-order curvature estimation.

---

> > ### Comment · Reviewer_mnxr · 2024-08-13
> >
> > I thank the authors for their rebuttal. The detailed response and the additional results addressed my concerns. I will keep the score.

---

### Official Review · Reviewer_Vex3 · 2024-07-20

**Soundness:** 3
**Presentation:** 3
**Contribution:** 3
**Rating:** 7
**Confidence:** 5

**Summary:**

The paper explores the use of loss curvature, under the hypothesis that test samples lie in high curvature regions, to improve the performance of membership inference attacks. It estimated the curvature by measuring the trace of the Hessian of the loss values. It demonstrates the viability of the approach in both white-box and black-box settings. It empirically validates the success of the approach on multiple datasets.

**Strengths:**

This paper proposes a simple but effective improvement over the canonical baseline of using loss estimation on train and test samples for membership inference attacks. It tests its effectiveness on three datasets and compares it with key baselines in the previous works. It also validates its effectiveness against differentially private models that provide provable privacy guarantees to a certain degree.

**Weaknesses:**

While the loss value based MIA evaluation generalizes to pretty much all class of models, it’s unclear if loss curvature based MIA attack would enable a similar generalization, i.e., going beyond the small-scale classification setup to self-supervised models, contrastive models, and other foundation models [1].

As fig. 6 itself highlights, the robustness of attack against a simple defense that subsamples the training data for shadow models is very low. Just using ½ of the training data renders the attack ineffective (albeit Carlini et al. suffers from the same limitation). Since shadow models are only a proxy of the true black-box model with training data information not publicly available, in practice they will likely only be trained on a subset of original training data of the block-box model, thus significantly reducing the attack success in black-box threat model.

1. Niu, Jun, Xiaoyan Zhu, Moxuan Zeng, Ge Zhang, Qingyang Zhao, Chunhui Huang, Yangming Zhang et al. "SoK: Comparing Different Membership Inference Attacks with a Comprehensive Benchmark." arXiv preprint arXiv:2307.06123 (2023).

**Questions:**

Why not use zero-order loss estimation in black-box method as it would be significantly more sample efficient than block-box curvature estimation.

Would using the first-order statistics, gradients of the loss w.r.t input, be as helpful as curvature in MIA attacks?

**Limitations:**

The paper proposes using second order statistics (loss curvature) over using loss values in membership attacks. However it will fail to generalize to networks that are piecewise linear, thus having zero curvature for all samples. A simple example of it would be the resnet models with convolution layer, ReLU activation, and Batchnorm normalization. While the paper does consider the resnet models, it doesn’t clarify the architecture uses. In the piecewise linear architecture, only the cross-entropy loss on output logits would contribute to loss curvature.

Thus it’s critical to ablate the choice of network architecture and demonstrate that despite the diversity in architectural designs, the approach succeeds in MIA attacks. A few simple variations would be changing the normalization layers, activation functions, and broadly the class of models (CNNs vs transformers).

---

> ### Author Rebuttal · Authors · 2024-08-06
>
> We thank the reviewer for their feedback, we address the weaknesses and questions below:
>
> 1.	While the loss value based MIA evaluation generalizes to pretty much all class of models, it’s unclear if loss curvature based MIA attack would enable a similar generalization, i.e., going beyond the small-scale classification setup to self-supervised models, contrastive models, and other foundation models [1].
>
> A:  The proposed curvature score is generic and only requires a valid loss function to measure model performance. Therefore, it can theoretically be applied to any machine learning model. The only limitation is with auto-regressive models, where defining input loss curvature is non-trivial. To test the generalizability of the method, we evaluated it across additional architectures. The results are presented in Table 3 in the rebuttal PDF. We appreciate the reviewer's input and will revise the paper to include these additional results.
>
> 2.	As fig. 6 itself highlights, the robustness of attack against a simple defense that subsamples the training data for shadow models is very low. Just using ½ of the training data renders the attack ineffective (albeit Carlini et al. suffers from the same limitation). Since shadow models are only a proxy of the true black-box model with training data information not publicly available, in practice they will likely only be trained on a subset of original training data of the block-box model, thus significantly reducing the attack success in black-box threat model.
>
> A:    Further investigation and code audit revealed that the performance degradation at 50% dataset size in Figure 6 was caused by a bug in the experiment. This one-line bug (see the diff in the rebuttal pdf) resulted in under-representing the membership inference attack (MIA) performance of our method and that of Carlini et al. After fixing this bug, the updated graph for Figure 6 in the rebuttal PDF (Figures 1 and 2 in the global response pdf) shows that the curvature-based method performs better than before. Specifically, as predicted by Theorem 4.3 beyond 20 − 30% (Figure 2 and 1 in the rebuttal pdf) subset Curv ZO LR matches or outperforms prior works in AUROC (better in TPR at low FPR) and Curv ZO NLL (Figure 1 in the rebuttal pdf) outperforms prior works beyond 10% subset size (compared to 40% previously). Both our method and Carlini et al.'s do not show a performance drop at 50% dataset size. We thank the reviewer for their input and will revise the paper to include these additional results.
>
> 3.	Why not use zero-order loss estimation in black-box method as it would be significantly more sample efficient than block-box curvature estimation.
>
> A:   This is indeed possible; however, our empirical results suggest curvature outperforms loss-based attack. We use the true loss (not estimated) for LOSS attack [as described Carlini et al.], which performs poorer than LIRA [Carilli et al] and our method (supported by Table 1 and Figure 4 in the main paper, and Table 2 in the rebuttal pdf). The main reason being loss-based MIA is non-parametric. Non-parametric MIA methods struggle at low FPR performance (also shown by Carlini et al.). Since the loss distribution is non-gaussian Carlini et al. suggest logit scaling to build parametric MIA i.e. LiRA. Our experimental results show that we outperform both of these methods (Carlini et al. and LOSS).
>
> 4.	Would using the first-order statistics, gradients of the loss w.r.t input, be as helpful as curvature in MIA attacks?
>
> A:  Indeed, first and zero-order statistics can be useful. To evaluate this, we compare these metrics to the memorization score from Feldman and Zhang, which is an independent metric of memorization in deep models. We find that curvature aligns best with memorization at the end of training and thus outperforms zero and first-order (i.e., loss and loss gradient) metrics. See the Table 4 in the rebuttal pdf. Table 4 shows the cosine similarity of Feldman Zhang (FZ) memorization scores (independent metric) with the different order statistics, for all the samples in CIFAR100 and top 5000 most memorized samples in CIFAR100 (according to Feldman and Zhang). Further, the loss-based attack [LOSS from Carlini et al.] is provided as reference in Table 2 (rebuttal pdf) which fails to outperform curvature. We thank the reviewer for their input and will revise the paper to include these additional results.
>
> 5.	However .... Thus it’s critical to ablate the choice of network architecture and demonstrate that despite the diversity in architectural designs, the approach succeeds in MIA attacks. A few simple variations would be changing the normalization layers, activation functions, and broadly the class of models (CNNs vs transformers).
>
> A: We consider the second derivative of the loss with respect to the input. Curvature estimation involves sampling the loss landscape at specific points to determine curvature. This method assumes that the function is smooth. If the function is piecewise linear, the zero-order (zo) method estimates the curvature of the smoothest function that intersects all sampled points, thereby smoothing the piecewise linear function. This approach is validated by our membership inference attack (MIA) results across various architectures, including ResNet18 with batch normalization and ReLU (as detailed in the main paper), and additional results for VGG11, VGG11_BN, Inception, and Vision Transformers (ViT-B16) as presented in the rebuttal document. The results demonstrate that zo curvature outperforms prior methods (see Table 3 in rebuttal PDF).
>
> Feldman, V. and Zhang, C. What neural networks memorize and why: Discovering the long tail via influence estimation. Advances in Neural Information Processing Systems, 33:2881–2891, 2020.
>
> N. Carlini, S. Chien, M. Nasr, S. Song, A. Terzis, and F. Tramer. Membership inference attacks from first principles. In 2022 IEEE Symposium on Security and Privacy (SP), pages 1897–1914. IEEE, 2022.

---

> > ### Comment · Reviewer_Vex3 · 2024-08-12
> > **Previous concerns addressed in the rebuttal**
> >
> > Thanks for posting a detailed rebuttal to the review. I believe that it had addressed the concerns raised in the review, thus I've increased my score to accept.

---

### Author Rebuttal · Authors · 2024-08-06

We thank all the reviewers for their input, there were a few common questions and weaknesses that we discuss and address below

1.	Performance of zero-order curvature estimation vs curvature.

A: We provide additional results on the effect of zero-order estimation. We run the proposed MIA attack on CIFAR100 and CIFAR10 datasets with zero order (ZO) estimation (black-box) and compare it against white-box Hutchinson curvature calculation [Garg et al.]. The results are provided in Table 1 and Figure 3 in the rebuttal pdf. Please note this comparison is a black-box (ZO) vs white-box (Hutchinson) comparison hence is provided only to illustrate the effect of ZO estimation.

2.	Improved MIA performance.

A: Further investigation and a code audit revealed that the performance degradation at the 50% dataset size in Figure 6 was due to a minor bug (see diff in Figure 4) in the experiment. This issue resulted in under-representing the membership inference attack (MIA) performance of our method and that of Carlini et al. This under-representation was in Figure 6 only, tables and other figures are unaffected. After correcting this bug, the updated graph for Figure 6 in the rebuttal PDF (Figures 1 and 2 in the global response PDF) shows that the curvature-based method performs significantly better than before. Specifically, as predicted by Theorem 4.3, beyond 20-30% (Figure 2 and 1 in the rebuttal pdf) subset, Curv ZO LR matches or outperforms prior works in AUROC and exhibits better TPR at low FPR, while Curv ZO NLL (Figure 1 in the rebuttal pdf) outperforms prior works beyond 10% subset size (compared to 40% previously). Neither our method nor Carlini et al.'s shows a performance drop at the 50% dataset size. We thank the reviewers for their input and will revise the paper to include and correctly represent these results, i.e. remove this specific weakness described in the paper.

3.	Performance on different architectures

A: We validated our membership inference attack (MIA) results across various architectures, including ResNet18 with batch normalization and ReLU (in the main paper), and additional results for VGG11 (without batch norm), VGG11_BN (with batch norm), Inception, and Vision Transformers (ViT-B16) as presented in the rebuttal pdf. The results demonstrate that ZO curvature outperforms prior methods (see Table 3 in rebuttal PDF) and is consistent with the takeaways in section 6.2 of the paper “Applications that demand high AUROC can use NLL approach, while those that demand high TPR at very low FPR should use the LR technique”

N. Carlini, S. Chien, M. Nasr, S. Song, A. Terzis, and F. Tramer. Membership inference attacks from first principles. In 2022 IEEE Symposium on Security and Privacy (SP), pages 1897–1914. IEEE, 2022.

I. Garg, D. Ravikumar, and K. Roy. Memorization through the lens of curvature of loss function around samples. arXiv preprint arXiv:2307.05831, 2023

---

### Decision · Program_Chairs · 2024-09-25

**Decision:**

Accept (spotlight)

**Comment:**

This papers studies how the second-order derivative of the loss w.r.t. the input data vector is revealing information about whether the data was used for training.

Reviewers generally like the work how it contributes to membership inference attack,  privacy auditing as well as that the paper provided both a theoretical analysis and experiments on a variety of settings.

All reviewers are supportive of the paper and I am happy to recommend accepting the paper to neurips.